CTRP3 attenuates inflammation, oxidative and cell death in cisplatin induced HK-2 cells

Zou Chenglin 1 2
Tang Xun 1
Guo Tingting 1
Jiang Tingting 3
Zhang Wenying 4
Zhang Jun gzh_zj@qq.com 1
1 Department of Nephrology, Zhujiang Hospital, Southern Medical University , Guangzhou , China
2 Department of Nephrology, The Second People’s Hospital of Jingzhou , Jingzhou , China
3 Department of Nephrology, The People’s Hospital of Guangxi Zhuang Autonomous Region , Nanning , China
4 The Second Affiliated Hospital of Guangzhou Medical University , Guangzhou , China
Navaneethabalakrishnan Shobana
Electronic publication date: 2023 Aug 23
Publication date: 2023
Volume: 11
Electronic Location ID: e15890
Received 2023 Feb 20; Accepted 2023 Jul 21
Copyright: ©2023 Zou et al.
Copyright year: 2023
Copyright holder: Zou et al.
License: This is an open access article distributed under the terms of the Creative Commons Attribution License, which permits unrestricted use, distribution, reproduction and adaptation in any medium and for any purpose provided that it is properly attributed. For attribution, the original author(s), title, publication source (PeerJ) and either DOI or URL of the article must be cited.
License URL: https://creativecommons.org/licenses/by/4.0/

Keywords: CTRP3, Cisplatin, Kidney, Injury

Funding: The authors received no funding for this work.

==============================
Cisplatin has been widely studied and found to be a highly effective anti-tumor drug. It has several side effects, including acute kidney injury (AKI). Cisplatin-induced AKI can be primarily attributed to oxidative stress, inflammation, and apoptosis. The CTRP3 adipokine is a new adipokine that exhibits antioxidant, anti-inflammatory, and antiapoptotic properties. Despite this, the role of CTRP3 in AKI remain unclear. In cisplatin-induced AKI models, our findings demonstrated that CTRP3 expression was decreased in human proximal tubule epithelial cells (HK-2). In the in vitro experiments, HK-2 cells were first transfected with an overexpression plasmid of CTRP3 (pcDNA-CTRP3) or a small interfering RNA for CTRP3 (si-CTRP3) and induced by cisplatin; and cell oxidative stress, inflammation, proliferation, and apoptosis were found to be present. Overexpressing CTRP3 inhibited oxidative stress through decreasing malondialdehyde (MDA) levels and increasing the activity of SOD and CAT. The mRNA levels of SOD1 and SOD2 were increased in response to CTRP3 overexpression. Additionally, CTRP3 decreased TNF-α and MCP-1 levels. Moreover, CTRP3 overexpression increased cisplatin-induced cell activity and decreased cell apoptosis, as indicated by the elevated numbers of EdU positive cells and decreased numbers of apoptotic cells. Consistent with these results, the overexpression of CTRP3 effectively elevated the mRNA levels of Bcl-2 and reduced the mRNA levels of Bax. In contrast, inhibition of CTRP3 expression by si-CTRP3 reversed the cisplatin-induced indices. Mechanistically, we found that the overexpression of CTRP3 can increase expression of Nrf2 and inhibit the activation of MAPK phosphorylation (ERK, JNK, and p38). Furthermore, inhibition of ERK, JNK and p38 activity eliminated aggravation of cisplatin-induced inflammation and apoptosis caused by CTRP3 knockdown. Additionally, the cisplatin-induced oxidative stress and activation of MAPK phosphorylation (ERK, JNK, and p38) in HK-2 cells were reversed by Nrf2 suppression by siRNA. Collectively, these results indicated that CTRP3 may identify as a novel target for AKI treatment and protect against cisplatin-induced AKI through the Nrf2/MAPK pathway.

Introduction

Acute kidney injury (AKI) is a serious clinical syndrome that affects many people and is characterized by tubular damage and sudden renal failure (Lameire et al., 2013). AKI has become a global health issue with high mortality and morbidity rates, with approximately 1.7 million deaths per year worldwide recorded (Mehta et al., 2015). The main causes of AKI are ischemia and cellular toxicity (Weidemann et al., 2008). Cisplatin is a common and effective chemotherapeutic drug used to treat human malignant tumours (Arany & Safirstein, 2003; Pabla & Dong, 2008). The most common cisplatin-associated nephrotoxicity is AKI, which affects approximately 25–30% of patients treated with cisplatin (Manohar & Leung, 2018), increasing the risk of developing chronic kidney disease (CKD) (Hsu, 2012). Cisplatin accumulates mostly in the kidney’s proximal tubular cells (Yao et al., 2007). Oxidative stress, inflammation, and cell apoptosis may contribute to cisplatin-induced AKI (Ozkok & Edelstein, 2014; Zhu et al., 2015). Oxidative stress plays an important role in cisplatin-induced AKI (Zhu et al., 2020b). ROS causes dysregulation MAPK signaling pathway (Ray, Huang & Tsuji, 2012), and excess ROS induce the MAPK signaling pathway and activate apoptosis (Checa & Aran, 2020). Research findings reported that nuclear factor erythroid 2-related factor (Nrf2) plays an important role in regulating oxidative stress (Li et al., 2018). Furthermore, studies have shown that the activation of Nrf2 pathway and the inhibition of MAPK pathway are involved in cisplatin-induced AKI (Jo et al., 2005; Sahin et al., 2010). However, the underlying mechanisms of cisplatin-induced AKI are complex and remain unclear (McSweeney et al., 2021).

C1q tumor necrosis factor-related protein-3 (CTRP3) belongs to a family of adiponectin paralogs called C1q/tumor necrosis factor-related proteins (CTRPs), which are highly conserved (Otani et al., 2015). Various tissues and organs, such as adipose tissues (Schäffler et al., 2003), skeletal muscles (Maeda et al., 2001), and liver (Peterson, Wei & Wong, 2010), express CTRP3. CTRP3 is expressed in vascular smooth muscle cells at the cellular level (Zhou et al., 2014). Recent studies have revealed that the expression level of CTRP3 decreases in the pathogenesis of acute injury in high glucose (HG)-induced H9C2 cells (Ma et al., 2017) and oxygen-glucose deprivation/reoxygenation (OGD/R)-induced hippocampal neurons (Ding, Wang & Song, 2021). Furthermore, several studies have shown that CTRP3 participated in neuroprotection and cardioprotection via reducing inflammatory response, apoptosis, and oxidative stress, suggesting that CTRP3 is a potentially protective factor (Gao, Qian & Wang, 2020; Ma et al., 2017). The increasing volume of research studies have suggested that CTRP3 is crucial in renal diseases, such as renal fibrosis (Chen et al., 2019), IgA nephropathy (Zhang et al., 2016) and diabetic nephropathy (Moradi et al., 2019). These findings demonstrated that CTRP3 overexpression improves renal fibrosis in the unilateral ureteral obstruction (UUO) rats and TGF-β1 induced tubular epithelial cells by the Notch signaling pathway (Chen et al., 2019) and inhibits polymeric IgA-stimulated human mesangial cells proliferation through the NF- κB signaling to attenuate the progression of IgAN (Zhang et al., 2016). Additionally, the expression level of CTRP3 decreases in HG-induced HK-2 and human glomerular mesangial cells, and overexpression of CTRP3 can inhibit cell proliferation and inflammatory response (Du et al., 2022; Hu, Li & Pan, 2019). Despite these findings, it is unknown how CTRP3 contributes to the AKI caused by cisplatin. Therefore, we speculate that CTRP3 may be involved in the pathogenesis of cisplatin induced acute kidney injury by mediating oxidative stress, inflammation and apoptosis. Recent research suggests that CTRP3 attenuated oxidative stress, inflammation and apoptosis, which were mediated by activation of Nrf2/HO-1 (Zhang & He, 2019) and suppressing p38 and JNK signaling pathways (Meng, Wang & Luo, 2019). Therefore, we hypothesize that CTRP3 is involved in cisplatin-induced kidney injury through Nrf2/MAPK signaling pathway. In this study, we discovered that activating CTRP3 greatly reduced the acute kidney damage caused by cisplatin, due to observations of improved oxidation, suppressed inflammatory expression, and decreased renal tubular apoptosis through the MAPK pathway. Furthermore, CTRP3-mediated oxidative stress and MAPK pathway activation were dependent on increased nuclear factor erythroid 2-related factor (Nrf2) expression, and the protective effects were absent in cells with Nrf2-knockdown. As a result, this study revealed that CTRP3 is crucial for AKI and may represent a possible therapeutic target for cisplatin-induced nephrotoxicity.

Materials and methods

Cisplatin-induced HK-2 cell model establishment

The human renal proximal tubular cell (HK-2) line was obtained from the American Type Culture Collection (ATCC). HK-2 cells were cultured in DMEM/F-12 medium (Gibco, Billings, MT, USA) ,which was supplemented with 10% fetal bovine serum (FBS), 100 U/mL penicillin, and 100 µg/mL streptomycin, then maintained in a humidified incubator at 37 °C in 5% CO2. Cells were planted in a 6-well culture plate for cisplatin (Qilu Pharmaceutical, Jinan, China) stimulation when confluence reached 80%.

Cell transfection

Expression plasmids containing CTRP3 cDNA (pcDNA-CTRP3) and negative controls (vector) were constructed by Jennio Biotech (Guangzhou, China). CTRP3 siRNA (siCTRP3) and negative control siRNA (siNC) were designed and synthesized by RiboBio (Guangzhou, China) and transfected to HK-2 cells using Lipofectamine 2000 (Thermo Scientific, Waltham, MA, USA) by following the manufacturer’s instructions. The cells were transfected for 24 h, and then used for further tests after being treated with or without cisplatin for 24 h. The siRNA targeting Nrf2 (siNrf2) and its negative control (siNC) were obtained from RiboBio (Guangzhou, China) and transfected into cells, following the manufacturer’s instructions.

RNA preparation and reverse transcription-quantitative polymerase chain reaction (RT-qPCR)

Total RNA was extracted from HK-2 cells using the TRIzol reagent (Invitrogen, Waltham, MA, USA, TR118-500) according to the manufacturer’s instructions. Using the M-MLV Reverse Transcriptase Kit (M1705; Promega, Madison, WI, USA), total RNA was reverse-transcribed to cDNA. Thereafter, quantitative PCR was performed using the GoTaq® qPCR Master Mix kit (A6002; Promega, Madison, WI, USA) on the quantitative PCR system (7300, Applied Biosystems, USA). β-actin was used as an internal control. The sequences of the primers used in qPCR were as follows (Table 1). The incubation and cycling conditions were 95 °C for 2 min, followed by 45 cycles of 95 °C for 15 s and 60 °C for 30 s. The relative mRNA expression levels were estimated using the 2−ΔΔCT method and normalized to that of β-actin. The experiment was repeated at least three times.

Table 1 Primers used in the study.

No.	Target gene	Sequence (sense, anti-sense, 5′–3′)	
1	CTRP3	GCTGCGTTTTACACCCTTTCTT
CGCCTTCACCGTTCCAGTTTTTA	
2	Bcl-2	CTGGGAGAACAGGGTACGATAA
TGGCTGGGAGGAGAAGATGC	
3	Bax	CAGGATGCGTCCACCAAGAA
CCTTGAGCACCAGTTTGCTG	
4	SOD1	CAGGGCACCATCTACTTCG
TCACCTTCAGCCAGTCCTTT	
5	SOD2	GCTGGAAGCCATCAAACG
TTAGAACAAGCGGCAATCTG	
6	β-actin	ATAGCATTTGGTTTAGTGGGTT
GGAGCCAATTATTTGTGAGCAT	

Western blot analysis

The protein levels were determined using a Western blot analysis. RIPA lysis mixture with 10% PMSF was used to prepare HK-2 cells lysates, and bicarbonic acid was used to measure the protein concentration. SDS-PAGE was used to separate an equivalent amount of protein, and the separated protein was then transferred to a PVDF membrane (Millipore, IPVH00010, Burlington, MA, USA). The membrane was incubated with primary antibodies after blocking for 1 h at room temperature with 5% non-fat powdered milk in PBS buffer, including against anti-CTRP3 (1:500, ab36870, Abcam; Cambridge, UK), anti-caspase-3 (1:1000, AF0722, Affinity, San Francisco, CA, USA), anti-p-JNK (1:1000, ab4821, Abcam), anti-JNK (1:1000, AF6318, Affinity), anti-p-ERK (1:1000, ab214036, Abcam), anti-ERK (1:1000, AF6240, Affinity), anti-p-p38 (1:1000, ab45381, Abcam), anti-p38 (1:1000, AF6456, Affinity), anti-Nrf2 (1:1000, AF0639, Affinity), anti-GAPDH (1:10000, KC-5G5, KANGCHENG, Guangzhou, China), and anti- β-actin (1:10000, KC-5A08, KANGCHENG) at 4 °C overnight. Subsequently, They were rinsed three times with a tris-buffered saline solution containing 0.1% Tween 20 (TBST) for five minutes each, then incubated with secondary antibody for one hour at room temperature. GAPDH and β-actin were used as internal controls. The bands were detected using an electrochemiluminescence (ECL) reagent (WBKLS0500, Millipore, Burlington, MA, USA); they were quantified using the Image J software (National Institutes of Health, Bethesda, MD, USA). Protein levels were normalized to that of GAPDH or β-actin. The experiment was repeated at least three times.

Measurement of oxidative stress markers and the levels of inflammatory factors

The activities of superoxide dismutase (SOD), catalase (CAT) and levels of MDA in the cisplatin-treated HK-2 cells were detected using commercial detection kits (Nanjing Jiancheng Bioengineer Institute, Nanjing, China). TNF-α and MCP-1 expression levels were measured in the culture supernatants of cisplatin-treated HK-2 cells using commercially available ELISA kits (Neobio Science, Shenzhen, China). The manufacturer’s instructions were followed for each step of the process. The samples were examined in triplicate.

Cell viability assay

The MTT (Sigma, St. Louis, MO, USA) assay was used to measure cell viability and was carried out in accordance with the manufacturer’s instructions. HK-2 cells were seeded in 96-well plates at a density of 1 × 104 cells/well, and 100 µl of DMEM complete medium (containing 10% FBS) was added to each well before incubating for 24 h. For the generation of formazan crystals, HK-2 cells were treated with 20 µL of MTT solution (0.5 mg/mL) and incubated for 4 h at 37 °C. 150 µL of dimethyl sulfoxide (DMSO) were added to the sample after the supernatant was removed in order to dissolve the crystals. Using an enzyme-labelling device, the absorbance was measured at 490 nm (Thermo Fisher, Waltham, MA, USA).

Ethynyl deoxyuridine (EdU) incorporation assay

Cell proliferation was measured using the EdU staining kit (Beyotime, Shanghai, China) as directed by the manufacturer. HK-2 cells were seeded in 24-well plates at a density of 1 ×105 cells per well. EdU reagents were added to the culture medium at a concentration of 20 µmol/L. They were incubated for 2 h before cells were fixed for 15 min at room temperature with 4% phosphate buffered paraformaldehyde and rinsed twice with phosphate-buffered saline (PBS). The nuclei were then stained with 1 mL of Hoechst 33342 (Sigma, USA) after the cells had been stained with 500 µL of fresh Click reaction solution. The cells were cultured for 10 min at room temperature in darkness. The percentage of EdU positive cells were calculated using a fluorescence microscope (Olympus, Tokyo, Japan).

Flow cytometry analysis of cell apoptosis

Cell apoptosis was measured using the Annexin V-FITC Apoptosis Detection Kit (Beyotime, Shanghai, China). HK-2 cells were seeded at a density of 1 × 105 cells per well in 6-well plates for 24 h before being treated with cisplatin (4 µg /ml) or DMEM. Cells were treated and digested with trypsin in the absence of ethylenediamine tetraacetic acid after treatment. The cells were then rinsed three times with cold PBS and resuspended in 100 µL of binding buffer. To resuspend cells, 5 µl of Annexin V-FITC staining fluid and 10 µl of propidium iodide staining fluid were added. A flow cytometer (BD Biosciences, Franklin Lakes, NJ, USA) was used to measure the apoptotic ratio of HK-2 cells, and the findings were analyzed using the Flow Jo software (TreeStar, Ashland, OR, USA).

TUNEL assay

In summary, apoptotic cells in HK-2 cells were detected using the TUNEL kit (Servicebio, Wuhan, China). HK-2 cells were seeded into a 24-well plate at a density of 5 × 104 cells/cm2 and incubated in TUNEL buffer at 37 °C for 2 h. Sections were stained with DAPI (Servicebio, Wuhan, China) after three washes with PBS. A microscope (NIKON, Tokyo, Japan) with a magnification of 100 × was used to count the total number of cells and the quantity of TUNEL-positive apoptotic cells in five distinct randomly selected high-power locations. The proportion of apoptotic cells was estimated by dividing the total number of cells by the number of TUNEL-positive cells.

Statistical analysis

Graphs were created using GraphPad Prism version 7.0 software (GraphPad, La Jolla, CA, USA). The data were analyzed using SPSS statistical software (version 19.0; IBM, Armonk, NY, USA), and the results are presented as mean ± SD. Data with regularly distributed groups were compared using the t-test, whereas data with more than two groups were compared using one-way ANOVA. The results of multiple groups were compared using ANOVA and LSD test. P-value < 0.05 was considered statistically significant.

Results

Expression of CTRP3 was lowly expressed in cisplatin-induced HK-2 cells

Cisplatin was used to stimulate the HK-2 cells to study the effect of CTRP3 in vitro. When cisplatin was administrated to HK-2 cells in different concentrations (0, 1, 2, 4, 8, 16, and 32 µg/ml) for 24 h, the viability of the cells noticeably reduced to 4 µg/ml (Fig. 1A). The treatment of 4 µg/ml cisplatin for 24 h was chosen as the condition for use in the next studies. RT-PCR and western blot analysis were used to ascertain the CTRP3 expression levels. In vitro study showed that cisplatin stimulation resulted in a significant decrease in CTRP3 mRNA expression level in HK-2 cells (Fig. 1B). Additionally, cisplatin stimulation caused a substantial reduction in the expression of CTRP3 protein (Fig. 1C).

Figure 1 Expression level of CTRP3 in cisplatin-induced HK-2 cells.

(A) HK-2 cells were treated with different concentrations of cisplatin for 24 h, and the cell viability was detected by the MTT assay. (B) The mRNA levels of CTRP3 were analyzed by RT-PCR. (C) Representative images of Western blotting and the quantitative analysis of CTRP3. (D) The transfection efficiency of pcDNA-CTRP3 was tested by RT-qPCR and Western blot. (E) The transfection efficiency of siCTRP3 was tested by RT-qPCR and Western blot. All experiments were repeated three times. Data are presented as mean ± SD (n = 3).

CTRP3 transfection efficiency was tested by RT-qPCR and Western blot analysis in HK-2 cells

To study the biological effect of CTRP3 on cisplatin-induced damage, HK-2 cells were transfected for 24 h with pcDNA-CTRP3 or si-CTRP3, followed by 24 h of cisplatin (4 µg/ml) stimulation. After that, the cells were collected for further examination. Transfection of pcDNA-CTRP3 was effective in increasing CTRP3 mRNA and protein levels, according to the results of RT-qPCR and Western blotting. (Fig. 1D), whereas CTRP3 mRNA and protein levels in HK-2 cells significantly decreased after transfection of CTRP3 siRNA (Fig. 1E).

Overexpression of CTRP3 attenuates cisplatin-induced oxidative stress response in HK-2 cells

Cisplatin-induced kidney damage is closely associated with oxidative stress (Dugbartey, Peppone & De Graaf, 2016). To investigate the role of CTRP3 in oxidative stress response, we evaluated the antioxidant capacity of HK-2 cells after CTRP3 overexpression and knockdown. The results show that after cisplatin treatment in HK-2 cells, superoxide dismutase (SOD) and CAT activity had dramatically reduced, which was followed by reducing the mRNA levels of SOD1 and SOD2 compared to the control group, whereas malondialdehyde (MDA) production was significantly enhanced. CTRP3 overexpression mitigated the cisplatin-induced inhibitory effect on SOD and CAT activity (Figs. 2A, 2C), as well as the mRNA levels of the SOD1 and SOD2 (Figs. 2D, 2E). While overexpression of CTRP3 prevented increased MDA production induced by cisplatin exposure (Fig. 2B). Furthermore, silencing CTRP3 resulted in the opposite effects (Figs. 3A–3E). Collectively, these findings demonstrated that overexpression of CTRP3 protects HK-2 cells against cisplatin-induced oxidative stress with decreased MDA levels and improved SOD and CAT activity.

Figure 2 Effect of CTRP3 overexpression on cisplatin-induced oxidative stress and inflammatory response in HK-2 cells.

(A) SOD activity. (B) The level of MDA. (C) CAT activity. (D) SOD1 mRNA level. (E) SOD2 mRNA level. (F) Level of TNF-α. (G) Level of MCP-1. (H) CTRP3 mRNA level. Data are presented as mean ± SD (n = 3).

Figure 3 Effect of CTRP3 silencing on cisplatin-induced oxidative stress and inflammatory response in HK-2 cells.

(A) SOD activity. (B) The level of MDA. (C) CAT activity. (D) SOD1 mRNA level.(E) SOD2 mRNA level. (F) Level of TNF-α. (G) Level of MCP-1. (H) CTRP3 mRNA level. Data are presented as mean ± SD (n = 3).

Overexpression of CTRP3 attenuates cisplatin-induced inflammatory response

Inflammation is a major feature of cisplatin-induced cytotoxicity (Pabla & Dong, 2008). Cisplatin-induced kidney damage is exacerbated by prolonged and excessive production of inflammatory cytokines and activation of inflammatory cells (Ozkok & Edelstein, 2014). In this research, we looked into how CTRP3 affected inflammation caused by cisplatin in the supernatant of HK-2 cells. MCP-1 and tumor necrosis factor-α (TNF-α) expression were induced by cisplatin. However, the elevations in the expression of MCP-1 and TNF-α were suppressed by treatment with pcDNA-CTRP3 plasmid (Figs. 2F, 2G). Contrarily, CTRP3 silencing aggravated cisplatin-induced inflammatory response in HK-2 cells was characterized by increased levels of MCP-1 and TNF-α (Figs. 3F, 3G). These findings suggested that overexpression of CTRP3 ameliorates cisplatin-induced inflammatory response by HK-2 cells.

Overexpression of CTRP3 improved cell viability and inhibited cisplatin-induced apoptosis

Cisplatin-induced kidney injury is mostly caused by apoptosis (Pabla & Dong, 2008). The MTT assay revealed a decreased cell viability after cisplatin treatment for 24 h (Fig. 4A). However, overexpression of CTRP3 improved the cisplatin-induced decrease in cell viability of HK-2 cells (Fig. 4A), whereas CTRP3 silencing promoted cisplatin-induced cell viability reduction (Fig. 5A). Meanwhile, anti-apoptotic protein Bcl-2 and pro-apoptotic protein Bax were also detected. Compared with the control group, the level of Bcl-2 mRNA was dramatically reduced in cisplatin treatment group, while the level of Bax mRNA was dramatically increased. Overexpression of CTRP3 effectively elevated the levels of Bcl-2 and reduced the levels of Bax induced by cisplatin (Figs. 4C, 4D). Contrarily, CTRP3 silencing aggravated the decrease of Bcl-2 level and promoted the increase of Bax level when compared with the cisplatin group (Figs. 5C, 5D). The EdU assay revealed that overexpression of CTRP3 reversed the inhibitory effect of cisplatin on HK-2 cell proliferation (Figs. 4E, 4F), whereas CTRP3 silencing promoted cisplatin-induced proliferation reduction (Figs. 5E, 5F). Furthermore, flow cytometry confirmed that cell apoptosis increased after cisplatin treatment. Overexpression of CTRP3 remarkably reduced HK-2 cell apoptosis (Figs. 4G, 4H), whereas CTRP3 silencing aggravated cisplatin-induced apoptosis (Figs. 5G, 5H). These results suggest that CTRP3 mediates cisplatin-induced apoptosis in HK-2 cells.

Figure 4 Effect of CTRP3 overexpression on cisplatin-induced cisplatin-induced proliferation and apoptosis.

(A) Cell viability was determined by the MTT assay after CTRP3 overexpression. (B) CTRP3 mRNA level. (C) Bcl-2 mRNA level. (D) Bax mRNA level. (E) Cell proliferation was determined by the EdU assay. (F) EdU-positive ratio of HK-2 cells. (G) Apoptosis was determined by flow cytometry. (H) Apoptotic ratio of HK-2 cells. Data are presented as mean ± SD (n = 3).

Figure 5 Effect of CTRP3 silencing on cisplatin-induced proliferation and apoptosis.

(A) Cell viability was determined by the MTT assay after CTRP3 silencing. (B) CTRP3 mRNA level. (C) Bcl-2 mRNA level. (D)Bax mRNA level. (E) Cell proliferation was determined by the EdU assay. (F) EdU-positive ratio of HK-2 cells. (G) Apoptosis was determined by flow cytometry. (H) Apoptotic ratio of HK-2 cells. Data are presented as mean ± SD (n = 3).

Renoprotective effect of CTRP3 overexpression on cisplatin-induced nephrotoxicity was associated with the inhibition of the MAPK pathway

The MAPK signaling pathways play an important role in regulating inflammation and apoptosis (Nath, 2006; Zmonarski et al., 2019). Previous studies have reported that MAPK pathways are activated in cisplatin-induced AKI, and inhibition of MAPK pathways alleviate cisplatin-induced inflammation and apoptosis (Sahu, Mahesh Kumar & Sistla, 2015). Furthermore, studies revealed that CTRP3 overexpression could alleviate inflammation and apoptosis by inhibiting MAPK pathway (Meng, Wang & Luo, 2019). Therefore, we speculate that CTRP3 may inhibit cisplatin-induced inflammation and apoptosis through MAPK pathways. Our results demonstrated that overexpression of CTRP3 can inhibit cisplatin-induced inflammation and apoptosis, to demonstrate that the renoprotective mechanism of CTRP3 in cisplatin-induced AKI, we investigated the effects of CTRP3 on the MAPK signaling pathways. We analyzed the levels of p38, JNK, and ERK phosphorylation in cultured HK-2 cells treated with cisplatin for 24 h by western blotting (Figs. 6A, 6B, 6C, 6D). The results showed that cisplatin increased the phosphorylation of p38, JNK, and ERK compared with control group, whereas CTRP3 overexpression significantly attenuated the phosphorylation of p38, JNK, and ERK in cisplatin-treated HK-2 cells. These results suggest that the protective effect of CTRP3 is mainly to inhibit the MAPK pathway.

Figure 6 Effect of CTRP3 overexpression on MAPK signaling pathway.

(A) Representative images of Western blotting of p-ERK, ERK, p-p38, p38, p-JNK, JNK and CTRP3. (B) The quantitative analysis of the protein levels of p-ERK. (C) The quantitative analysis of the protein levels of p-JNK. (D) The quantitative analysis of the protein levels of p-p38. (E) The quantitative analysis of the protein levels of CTRP3.Data are presented as mean ± SD. (n = 3).

Inhibition of MAPK reversed cisplatin-induced apoptosis and inflammatory response

To verify the protective effect of CTRP3 on cisplatin-induced kidney injury by inhibiting the MAPK signaling pathway, we investigated the effect of the MAPK pathway on inflammatory response and apoptosis. First, 50 nmol of p38 inhibitor, SB203580 (MedChem Express, Monmouth Park, NJ, USA), 40 nmol of JNK inhibitor, SP600125 (MedChem Express, Monmouth Park, NJ, USA), and 72 nmol of ERK inhibitor, U0126 (MedChem Express, Monmouth Park, NJ, USA) were added to the cisplatin-treated cells to interfere the activation of the MAPK pathway. After cisplatin treatment, MCP-1 and TNF-α levels increased significantly in HK-2 cells with CTRP3 knockdown. SB203580, SP600125, and U0126 pre-treatments evidently attenuated siCTRP3-aggravated expression levels of MCP-1 and TNF-α in HK-2 cells after cisplatin exposure (Figs. 7F, 7G). Additionally, the rate of apoptosis was tested through TUNEL staining and the apoptosis related protein caspase three cleavage was detected using western blot. The results showed that CTRP3 knockdown increased TUNEL-positive cells and caspase-3 cleavage in cisplatin-induced HK-2 cells. However, these effects were significantly reversed by the inhibition of p38, JNK and ERK with SB203580, SP600125, and U0126, respectively (Figs. 7A, 7B and Figs. 7C, 7D). However, there was no difference of MCP-1, TNF-α and cleaved-caspase 3 level and TUNEL-positive cells between these three groups pretreatment with MAPKs inhibitors. These findings suggest that CTRP3 inhibited the MAPK signaling pathway, which attenuated cisplatin-induced inflammation and apoptosis.

Figure 7 Effect of the inhibitors of MAPK pathway on cisplatin-induced inflammation and apoptosis.

(A) Representative images of TUNEL staining (100 ×magnification, scale bar, 50 mm). (B) The number of TUNEL-positive cells were counted. (C) Representative images of Western blotting of pro-caspase 3, cleaved caspase 3 and CTRP3. (D) Quantification of cleaved caspase-3/pro-caspase 3 ratio. (E) The quantitative analysis of the protein levels of CTRP3. (F) Level of MCP-1. (G) Level of TNF-α. Data are represented as mean ± SD (n = 3).

Nrf2 knockdown reversed CTRP3-mediated oxidative stress and MAPK pathway activation in cisplatin-stimulated HK-2 cells

Nrf2 signaling pathway regulates the production and clearance of cellular ROS (Deng et al., 2019). Moreover, evidence indicates that Nrf2 play crucial roles in cisplatin-induced renal toxicity (Shelton, Park & Copple, 2013). CTRP3 overexpression could inhibit oxidative stress by enhancing Nrf2 activation (Zhang & He, 2019). Therefore, we speculated that CTRP3 prevents cisplatin-induced oxidative stress by activating Nrf2. The findings showed that compared with the control group, the level of Nrf2 protein in HK-2 cells was significantly reduced in cisplatin treatment group (Figs. 8A, 8B). Furthermore, compared with the cisplatin+pcDNA-CTRP3 group, CTRP3 overexpression-enhanced SOD and CAT activity were significantly reduced by Nrf2 silencing (Figs. 9A, 9C), and the result was consistent with the levels of SOD1 and SOD2 mRNA (Figs. 9D, 9E). Additionally, Nrf2 knockdown effectively reversed the repressive properties of CTRP3 overexpression on MDA production (Fig. 9B). These results suggest that CTRP3 mediates cisplatin-induced oxidative stress through activating Nrf2.Previous studies have shown that oxidative stress could induce cell apoptosis by activating MAPK signaling (Guo et al., 2018). Our findings showed that CTRP3 inhibited cisplatin-induced apoptosis and inflammatory through the MAPK signaling pathway. Therefore, we further investigated the relationship between Nrf2 activation and MAPK pathways. The results demonstrated that CTRP3 overexpression significantly attenuated the phosphorylation of p38, JNK, and ERK in cisplatin-treated HK-2 cells .However, these effects were markedly reversed by Nrf2 knockdown (Figs. 8C, 8D, 8E). These results suggest that CTRP3 mediates oxidative stress through Nrf2 to regulate the MAPK pathway.

Figure 8 Effect of siNrf2 on MAPK signaling pathway.

(A) Representative images of Western blotting of p-ERK, ERK, p-p38, p38, p-JNK, JNK, Nrf2 and CTRP3. (B) The quantitative analysis of the protein levels of Nrf2. (C) The quantitative analysis of the protein levels of p-ERK. (D) The quantitative analysis of the protein levels of p-JNK. (E) The quantitative analysis of the protein levels of p-p38. (F) The quantitative analysis of the protein levels of CTRP3. Data are represented as mean ± SD (n = 3).

Figure 9 Effect of siNrf2 on cisplatin-induced oxidative stress in HK-2 cells.

(A) SOD activity. (B) The level of MDA. (C) CAT activity. (D) SOD1 mRNA level. (E) SOD2 mRNA level. (F) CTRP3 mRNA level. Data are represented as mean ± SD (n = 3).

Discussion

Consistent with findings on the expression of CTRP3 in heart failure (Gao et al., 2019) and animals with myocardial ischemia (Yi et al., 2012), our findings showed that expression levels of CTRP3 mRNA and protein were significantly reduced in cisplatin-treated HK-2 cells. In response to cisplatin stimulation, overexpression of CTRP3 increased the viability of HK-2 cells and reduced the oxidative stress, inflammation, and apoptosis that cisplatin causes. Furthermore, the Nrf2/MAPK signaling pathway was involved in renal protective effects.

Cisplatin is primarily eliminated through the kidneys, and epithelial cells of the renal tubules have a significantly higher concentration of cisplatin compared to that in the blood. The primary factor that contributes to cisplatin-induced kidney damage is its high concentration (Holditch et al., 2019). Inflammation, oxidative stress and tubular apoptosis have been implicated in the pathogenesis of cisplatin-induced AKI (Fang et al., 2021). Oxidative stress leads to the increase of MDA level and the decrease of SOD activity (Ruiz et al., 2013), which are crucial in the kidney damage caused by cisplatin (Manohar & Leung, 2018). Previous studies have shown that CTRP3 overexpression can inhibit oxidative stress induced by HG and OGD/R with decreased level of malondialdehyde (MDA) and increased SOD activity (Song et al., 2022; Zhang & He, 2019). Consistent with these results, our research demonstrated that CTRP3 overexpression increased the SOD and CAT activity and decreased MDA level. SODs, which include SOD1, SOD2, and SOD3, are peroxide catalysts that may convert ROS into hydrogen peroxide and then into water, preventing the generation of nitrogen oxides and other hazardous byproducts (Zelko, Mariani & Folz, 2002). We found that CTRP3 overexpression significantly increased the levels of SOD1 and SOD2 mRNA after cisplatin treatment. Inflammation is thought to play a role in the pathogenesis of AKI. Sustained and excessive production of inflammatory cytokine and activation of inflammatory cells aggravate cisplatin-induced kidney damage (Ozkok & Edelstein, 2014). According to our study findings, which were consistent with findings from a study suggesting that CTRP3 overexpression inhibited uric acid-induced inflammation with decreased levels of TNF-α and IL-6 (Zhang et al., 2022), HK-2 cells exposed to cisplatin had lower levels of TNF-α and MCP-1 after treatment with CTRP3. Additionally, we found that CTRP3 overexpression alleviated the effects of cisplatin on the viability and apoptosis of HK-2 cells, consistently, the levels of apoptosis-related proteins Bcl-2 and Bax were reversed, whereas CTRP3 silencing aggravated these effects. These data indicated that CTRP3 overexpression protected against cisplatin-induced apoptosis. Several studies indicated that CTRP3 possesses anti-apoptosis effects. Song et al. (2020) demonstrated that CTRP3 overexpression promoted cell proliferation and reduced apoptosis of AC16 cardiomyocytes treated with HG. Overexpression of CTRP3 attenuated the OGD/R-caused apoptosis with increased Bcl-2 expression and decreased Bax expression (Ding, Wang & Song, 2021). These results were consistent with our findings.

Next, in this study, we further investigated the mechanisms underlying the antioxidant, anti-inflammatory, and anti-apoptotic effects of CTRP3. Cisplatin-induced ROS activates a variety of signaling pathways including MAPKs and aggravate its toxicity (Sung et al., 2008), and the progression of acute kidney impairment caused by cisplatin was found to be closely related with the MAPK signaling pathway. Members of the MAPK family such as ERK, JNK, and p38 are activated after cisplatin treatment and contribute to renal cell death (Pabla & Dong, 2008). MAPK inhibition reduces cisplatin-induced kidney injury (Jo et al., 2005). Furthermore, CTRP3 has been shown to suppress p38 MAPK, thereby alleviating cardiac hypertrophy (Zhang et al., 2019). CTRP3 could also reduce inflammation by inhibiting ERK1/2 and P38 MAPK pathways in lipopolysaccharide-induced monocytes (Zhu et al., 2020a). Consistent with these studies, our study revealed that cisplatin administration significantly increased the phosphorylation of p38, JNK, and ERK, whereas overexpression of CTRP3 reversed these cisplatin-induced indices. Treatment with SB203580, SP600125, and U0126 remarkably reduced the effect of si-CTRP3 on cisplatin-induced inflammation and apoptosis, as indicated by the decreased levels of TNF-α and MCP-1 and reduced numbers of TUNEL-positive cells. In addition, the promotion of caspase-3 cleavage caused by CTRP3 silencing was reversed by these inhibitors. As a result, the renoprotective effect of CTRP3 overexpression against cisplatin nephrotoxicity was related to the inhibition of MAPK pathway.

The Nrf2 is an essential transcription factor that regulates cellular oxidative stress. It is crucial for maintaining intracellular oxidative homeostasis by causing the regulation of a number of antioxidant proteins (Kumar et al., 2014). Furthermore, several studies have demonstrated that the Nrf2 signaling pathway mediated the protection of cisplatin-induced AKI (Sahin et al., 2010). Consistent with Ding, Wang & Song (2021), our study verified that Nrf2 expression is significantly decreased in HK-2 cells, whereas CTRP3 overexpression significantly upregulated Nrf2 expression. Furthermore, CTRP3 increased Nrf2 activity in ARPE-19 cells stimulated by high glucose (Zhang & He, 2019). Several studies have discovered that excessive ROS act as intracellular signaling molecules, activating MAPK signaling (Fan et al., 2020), whereas Nrf2 activators alleviate cisplatin-induced ROS generation (Nathan, 2003). Previous studies have shown that CTRP3 overexpression could inhibit oxidative stress by enhancing Nrf2 activation in HG-stimulated ARPE-19 cells, which was accompanied with increased SOD activity and decreased levels of MDA (Zhang & He, 2019). Consistent with these findings, our results showed that the effects of CTRP3 overexpression on SOD and CAT activity and MDA production were significantly reversed by Nrf2 knockdown. Furthermore, Nrf2 silencing inhibited the increase of SOD1 and SOD2 mRNA levels caused by CTRP3 overexpression. We have found that CTRP3 could inhibit cisplatin-induced oxidative stress by increasing the activity of Nrf2, and reduce cisplatin-induced inflammation and apoptosis by inhibiting MAPK pathway. Then, whether there is a relationship between Nrf2 and MAPK pathway. Several studies have reported that Nrf2 activity can regulate the activation of the MAPK signaling pathway. Adenovirus-induced Nrf2 overexpression was found to significantly reduce ERK1/2 activation (Tan et al., 2011). Contrarily, the loss of Nrf2 function resulted in the activation of MAPK signaling and increased inflammation (Jiang et al., 2014). Consistent with these studies, our study showed that Nrf2 silencing could reverse the inhibitory effects of CTRP3 overexpression on the phosphorylation of p38, JNK, and ERK.

This study had several limitations. We mainly examined the impact of CTRP3 on cisplatin-induced HK-2 cells in vitro. Future research will take into account an in vivo animal studies.

Conclusion

CTRP3 overexpression may be a novel strategy for preventing the adverse effects of cisplatin, which are mainly due to oxidative damage, inflammation, and apoptosis. This finding was partially due to CTRP3′s ability to mediate the Nrf2/MAPK pathway. Therefore, CTRP3 may be a novel target for preventing cisplatin-induced kidney injury.

Supplemental Information

Supplemental Information 1 CTRP3 mRNA expression levels.

Click here for additional data file.

Supplemental Information 2 Raw data of CAT, SOD1, mRNA, SOD2, mRNA, and apoptosis-related proteins Bcl-2 and Bax were applied for data analyses

Used in preparation for Fig. 2C, D, E, Fig. 3C, D, E, Fig. 4C, D, Fig. 5C, D, and Fig. 9C, D, E.

Click here for additional data file.

Supplemental Information 3 WB gels.

Click here for additional data file.

Supplemental Information 4 Raw data of cell viability, oxidative stress, inflammation, apoptosis and MAPK pathway related proteins were applied for data analyses

Used in preparation for Fig. 1, Fig. 2A, B, F, G, Fig. 3A, B, F, G, Fig. 4A, F, H, Fig. 5A, F, H,, Fig. 6B, C, D, Fig. 7B, F, G, Fig. 8B, C, D, E, and Fig. 9A, B.

Click here for additional data file.

Additional Information and Declarations

Competing Interests

Author Contributions

Data Deposition

The authors declare that there are no competing interests.

Chenglin Zou conceived and designed the experiments, performed the experiments, analyzed the data, prepared figures and/or tables, authored or reviewed drafts of the article, and approved the final draft.

Xun Tang conceived and designed the experiments, analyzed the data, prepared figures and/or tables, and approved the final draft.

Tingting Guo performed the experiments, analyzed the data, prepared figures and/or tables, and approved the final draft.

Tingting Jiang performed the experiments, analyzed the data, prepared figures and/or tables, and approved the final draft.

Wenying Zhang analyzed the data, prepared figures and/or tables, and approved the final draft.

Jun Zhang conceived and designed the experiments, analyzed the data, prepared figures and/or tables, authored or reviewed drafts of the article, and approved the final draft.

The following information was supplied regarding data availability:

The raw measurements are available in the Supplemental Files.

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
