# Peer review of "CTRP3 attenuates inflammation, oxidative and cell death in cisplatin induced HK-2 cells"

_PeerJ, doi:10.7717/peerj.15890_

## Round 0.1 · original submission · Major Revisions

The manuscript has been assessed by three independent reviewers and I strongly suggest addressing the concerns raised by all three reviewers before your paper can be considered for publication. The authors need to provide enough background information and clear rationale behind the study. More concentration is required on linking different experiments and reason for performing those to bring a flow while reading manuscript. Figures need to be improved in terms of representation and labeling. Extensive revision of experimental design, results and discussion section is highly recommended.

·

Basic reporting

The current manuscript focuses on the role of CTRP3 in HK-2 cell line-induced oxidative stress, inflammation, and cell death. Although the study was well-designed, certain adjustments must be made.

The introduction can be expanded upon since it seems to contain little information. The introduction might use more details about earlier works.

Experimental design

No comment

Validity of the findings

This is major revision which need to be implemented. Kindly provide the proper legend for each figure. None of the figures had p-values indicated. Please do that. Just writing in the legend does not indicate whether a change is significant or not and it is hard to read. Moreover, in the rearrangement of the graphs the p-values should be presented in comparison to the control.

It is difficult to tell which group is significant to which group. For instance, Fig. 1 D: Control and Cisplatin+pcDNA-CTRP3 don't appear to differ considerably, but the figure seems to show that they do as control doesn't have #. Please reposition the figure. Authors may use the alphabet as the same alphabet will be considered in the same group.

Additional comments

As Bax and BCL2 are key regulators of apoptosis, the author can check additional mRNA levels of Bax, BCL2, SOD1, SOD2, and catalase to strengthen their data.

The author has already measured the level of SOD, and the mRNA levels of SOD1 and SOD2 will provide additional support for the data. SOD and catalase work in tandem, with SOD converting superoxide to peroxide and catalase converting peroxide to water and oxygen. in that context, Catalase activity can be measured in addition to SOD and MDA levels.

Reviewer 2 ·

Basic reporting

English has to be improved in certain places for better understanding of the readers.

a) The spelling of "apoptosiss" in line 18 has to be checked.

Experimental design

No comments

Validity of the findings

No comment.

Additional comments

The authors have done a commendable job by exploring the relationship between CTRP3 protein and Cisplatin, an anti-tumor drug. However, I would like the following issues to be addressed before accepting this research work for better understanding of the readers.

i) Please explain line 183. It seems that the authors are measuring the number of viable cells in ug. Please mention the number of viable cells as compared to control.
ii) Figure 5 A and 6 A containing the western blots has to explained better.

Reviewer 3 ·

Basic reporting

The authors have shown that the CTRP3 protein and NrF2 pathway are important in Cisplatin induced cell death and possibly in AKI. The manuscript includes series of experiment in HK-2 cells to evaluate the CTRP3 and Nrf2 protein upon Cisplatin treatment. However, the manuscript is poorly designed and written in a hasty manner. Authors have left a wide scope to improve the manuscript for better language and scientific argumentation.
The introduction part lacks a clear argument on how CTRP3 is connected to Cisplatin impact on AKI. The authors should elaborate on mechanisms and literature arguing how CTRP3 role might be regulated by Cisplatin during AKI.

Experimental design

For manuscript experiments, please include following corrections.

1. Include density values for corresponding proteins band in immunoblot images.

2. Results 3.5 states contradictory conclusions saying “CTRP3 mediates cisplatin-induced apoptosis in HK-2 cells” however, in the results (figure A) show higher cell viability with CTRP3 presence and figure (D) displays lesser apoptosis in the absence.
3. In result 3.7, authors directly initiating the evaluation of MAPK pathway without any rationale and without evaluating other relevant pathways i.e. NF-kB pathway. Author should include a logical reasoning on what basis MAPK pathway analysis was performed.
4. Author should also include data from NF-kb pathway to confirm the MAPK specificity. Analysis for pro-apoptotic (Bax/Bak) and survival (Bcl-2, Bcl2-A1) proteins should be included to strengthen the claim for cell viability with CTRP3 overexpression.

5. In figure 4E, caspase-3 immunoblots only display a single band and no cleaved form is shown as author claims in result 3.8. Please label the correct molecular weight of caspase 3 and include cleaved form to confirm activation.

6. Again, in result 3.9, author directly begins with Nrf2 specific role without any rationale to previous results. Please provide a stepwise logical explanation for choosing NrF2 for analysis. Author should correct the last line, use ‘activation or phosphorylation’ instead of activation of p-38, p-JNK and P-ERK.

Validity of the findings

Overall, the experimental results support the manuscript title and proposed questions. However, the manuscript is not very clear in explaining the hypothesis and results therefore needs to be improved at a greater extent in experimental designs and language.

Additional comments

Authors should explain results and conclusions with clarity and rationale. The manuscript lacks the connections and flow of information one results.

---

## Round 0.2 · Minor Revisions

The authors have done a great job by improving the overall quality of the manuscript. However, there are few concerns to be addressed before considering for the publication. CTRP3 expression levels needs more validation since the study is centered around the same. The authors are suggested to provide the full and cleaved form of caspase 3 in the same blot.

·

Basic reporting

More information regarding previous works by the authors has been included as suggested and the content is sufficient to illustrate the rationale behind the paper's objective.

Experimental design

No Comments

Validity of the findings

The authors have addressed the question raised throughout the revision process and explained it. Additional experiments were also carried out by the authors to support their research conclusions.

Additional comments

As a result, every question has been answered. I have no further doubts and recommend that the paper be approved.

Reviewer 2 ·

Basic reporting

No comments

Experimental design

No comments

Validity of the findings

No comments

Additional comments

The authors have made the necessary changes suggested previously. The manuscript is ready to be published.

Reviewer 3 ·

Basic reporting

The authors have incorporated many revision suggestions and executed significant efforts to improve the manuscript. However, to maintain scientific rigor for the manuscript, the following corrections must be included.

Experimental design

Please include the following major corrections.

1. Since the manuscript centered on the role of CTRP3 in HK-2 cells upon cisplastin treatment, the status of CTRP3 expression is crucial to confirm the cisplastin-mediated effect in HK-2 cells. Additionally, CTRP3 knockdown and overexpression were performed in the transient manner (not the stable expression system) it becomes necessary to confirm the targeted gene/protein every time in all observations (because of unavoidable transfection efficiency variability). Therefore, Fig. 6 (A), Fig. 7 (C), and Fig. 8 (A) must include CTRP3 protein levels in immunoblots.

2. In Fig. 3, 4, 5, and 8, mRNA expression data should also include the expression levels of CTRP3.

3. In the case of caspase-3 activation, results should include both full caspase 3 and cleaved caspase 3 in a single immunoblots strip. Displaying either of can single band is not sufficient for showing apoptosis-specific results. Please refer to DOI: 10.1126/sciadv.aau9433 manuscript for representation.

Minor points-

1. Figures 4 and 5, in representative FACS plots, show the frequencies in visible fonts (bigger and bold).

2. Show density values for corresponding proteins band in representative immunoblot images and individual data points in quantitative analysis.

Validity of the findings

CTRP3 expression status is warranted in several performed experiments to confirm the validity of results.

---

## Round 0.3 · accepted · Accept

The authors have adequately addressed the comments by the reviewers and the manuscript can be accepted for publication.